# Remodeling of Germ Cell mRNPs for Translational Control

**DOI:** 10.3390/biology14101430

**Published:** 2025-10-17

**Authors:** Brett D. Keiper, Hayden P. Huggins

**Affiliations:** 1Department of Biochemistry and Molecular Biology, Brody School of Medicine at East Carolina University, Greenville, NC 27834, USA; 2Biotechnology Program (BIT), Jordan Hall, North Carolina State University, Raleigh, NC 27695, USA; hphuggin@ncsu.edu

**Keywords:** mRNA translational control, messenger ribonucleoproteins (mRNPs), eukaryotic translation initiation factor 4E (eIF4E), eIF4E-interacting proteins (4EIPs), liquid–liquid phase separation (LLPS), biomolecular condensates, germ cell granules, ribosomes, macrostructures, substructures

## Abstract

**Simple Summary:**

Translational control is a critical mode of gene expression in cells unable to suitably modulate transcription, such as germ cells and neurons. Here we feature new insights into the unexpected properties of messenger ribonucleoprotein (mRNP) complexes that carry regulated mRNAs in these cells. Recently, there has been much attention on large germ granules, to which the mRNPs partition in the germline. Such granules are dynamic non-membrane-bound liquid–liquid phase-separated (LLPS) condensates. Ribosomes are excluded, indicating that translation is possible only outside of or at the periphery of these condensates. Interestingly, mRNA cap-binding eIF4E paralogs and their binding partners, which regulate translation initiation, take up defined positions in these condensates. Recent findings suggest that the condensates may be organizational but not functional for mRNA regulation. It is possible they result from the regulation imposed upon the mRNPs. We now know far more about translational repression by mRNPs and the anatomy of the germ cell condensates than we do about how they work in concert. Their integration is essential for synthesizing new proteins in the right time and place during development. One thing appears certain: we need a new concept of mRNA–protein dynamics in cells undergoing rapid change.

**Abstract:**

The localization and remodeling of mRNPs is inextricably linked to translational control. In recent years there has been great progress in the field of mRNA translational control due to the characterization of the proteins and small RNAs that compose mRNPs. But our initial assumptions about the physical nature and participation of germ cell granules/condensates in mRNA regulation may have been misguided. These “granules” were found to be non-membrane-bound liquid–liquid phase-separated (LLPS) condensates that form around proteins with intrinsically disordered regions (IDRs) and RNA. Their macrostructures are dynamic as germ cells differentiate into gametes and subsequently join to form embryos. In addition, they segregate translation-repressing RNA-binding proteins (RBPs), selected eIF4 initiation factors, Vasa/GLH-1 and other helicases, several Argonautes and their associated small RNAs, and frequently components of P bodies and stress granules (SGs). Condensate movement, separation, fusion, and dissolution were long conjectured to mediate the translational control of mRNAs residing in contained mRNPs. New high-resolution microscopy and tagging techniques identified order in their organization, showing the segregation of similar mRNAs and the stratification of proteins into distinct mRNPs. Functional transitions from repression to activation seem to corelate with the overt granule dynamics. Yet increasing evidence suggests that the resident mRNPs, and not the macroscopic condensates, exert the bulk of the regulation.

## 1. Introduction

The post-transcriptional regulation of gene expression is vital for animal cells undergoing differentiation or any purposeful change. Regulation takes the form of mRNA translational control in reproduction to differentiate germline stem cells to eggs/oocytes, sperm, and eventually embryos [1,2,3,4,5], as well as for neuronal differentiation, learning, and even in autism [6,7,8,9]. These cells achieve remarkable new fates without substantial transcriptional activation [10,11]. The decoration of mRNAs with trans-acting factors like RBPs and miRNAs occurs as they form ribonuclear protein (mRNP) complexes that exert translational repression and/or mRNA destabilization [12,13,14,15]. Such mRNPs are consequential for cells that must exercise some memory for their protein synthesis program [6,16,17]. Innovations in transcriptomics and proteomics indicate that these specialized cell types generally utilize mRNA regulation and protein turnover mechanisms to supersede transcriptional regulation. These become the primary regulatory means for germline, embryo, and neuronal development/learning [18]. In germ cells, these mRNPs traffic through perinuclear LLPS condensates (often referred to as granules) that seem to characterize the “non-somatic” nature of the germ lineage, especially in maintaining its transgenerational totipotency [19,20]. Yet they appear not to be functionally determinant for mRNA regulation or fertility per se [21]. The work on neuronal mRNP translational control has been reviewed elsewhere [9,22]. Similarly, numerous reviews provide a more comprehensive history of germ cell and embryo RBPs, mRNA repression, small RNA regulation (miRNAs, endo-siRNAs, etc.), cytoplasmic poly(A) elongation, and mRNA turnover (reviewed in [5,14,15,17,23,24,25,26,27,28,29]). This review instead focuses on the contemporary thought surrounding how mRNPs are regulated in condensates as this relates to their recruitment to ribosomes by eIF4 factors in animal germ cells and embryos.

## 2. Topical Review

### 2.1. History of Germ Granules and mRNPs: Microenvironments to Sort, Decorate, and Repress mRNAs

Originally named P granules in *C. elegans*, the condensates were originally observed to segregate with the P lineage (germline) during embryogenesis and through adult gonad development [30,31]. It was recognized that the granules recruit mRNAs but lack ribosomal RNA [32,33]. They also accumulate RNA-binding proteins, such as PGL-1, OMA-1, IFET-1, GLD proteins, GLS-1, etc., that are involved in sequestration, translational repression, polyadenylation, and mRNA decay [34,35,36,37,38,39,40]. Some of the proteins are sequence-specific, binding particular sequence motifs for the direct control of translation, both in and outside of condensates. Such proteins nucleate a small-scale RNA–protein complex, defined for the sake of this review as an mRNP. The “macro-scale” condensates (a few hundred nanometers to several micrometers) organize large arrays of these mRNPs and have dynamic characteristics that make them appear “liquid-like”, promoting the idea that they may shuttle or swap mRNPs between various complexes ([41,42] and Figure 1). More recently it has been recognized that such condensates can act to organize biochemical functions or pathways in cells [43]. Together these attributes suggest that considerable remodeling of mRNP structures on mRNAs may take place in germ cell condensates.

Given their perinuclear position and protein/RNA components, germ granules seem a likely repository of repressed mRNAs. Many germ cell mRNPs localize to LLPS condensates that associate with the nuclear pore complex (NPC) [41,44,45,46,47]. The assembly of RBPs and small RNAs onto these mRNAs is thought to occur as they traffic through the granules as an essential mode of mRNP remodeling [14,15,33]. Germ granules are enriched in small-RNA-processing complexes, as well as Argonaute-associated RNA-induced silencing complexes (RISC) [15,42,48,49]. In addition, several P body components, such as CGH-1/DDX6 and CAR-1/LSM14, are found in germ granules adjacent to small-RNA-silencing factors, such as the Argonaute WAGO-4 [50,51,52]. The evidence, therefore, indicates that preparation for substantial mRNA turnover or even licensing may also be accomplished in these condensates.

However, the eventual fate of many condensate-enriched mRNAs is translation. It has long been known that pools of stored germline mRNAs are recruited to ribosomes from mRNP complexes like those described above, some of which reside within germ granules during long inert periods [12,13,33,53,54]. The way mRNAs are handled as they leave the nucleus sets the agenda for their translational control in the cytoplasm [24,55]. Observations of mRNA localization and the coincident translational repression of *Nanos* and *pos-1* mRNAs in flies, zebrafish, and worms are consistent with the acquisition of repressive RBPs in the granules [32,56,57,58,59,60,61,62]. However, other mRNAs like *mex-3* and *glp-1* are translationally regulated by mRNPs that are likely free from such condensates [27,63]. Interestingly, Nanos and POS-1 proteins participate in such cytoplasmic translational repression following the de-repression of their own mRNAs [64,65,66,67]. Furthermore, the physical composition of the cytoplasm is non-homogeneous (e.g., distribution of proteins, yolk, mitochondria, ribosomes, etc.), especially in early embryos that must establish polarity for future developmental axes. Therefore, the landscape of the cytoplasm outside of granules may further influence spatiotemporal mRNA translational control in dividing blastomeres for cell-specific protein synthesis [59,63,68,69,70].

### 2.2. Translation on the Border: Ribosomes on the Periphery of Germ Cell Condensates

As early as 1971, electron micrographs of *Drosophila* germ plasm from eggs and embryos showed electron-dense bodies (eventually called germ granules) that were surrounded by tiny particles thought to be ribosomes [71]. Since then it has been postulated that mRNA must emerge from these germ granules to participate in active translation. Indeed, nearly all subsequent studies on animal germ cells and embryos seemed to uphold this simple “mRNA release” model to engage in translation (reviewed in [15,45,72,73,74,75]). It has been argued that the size exclusion by the germ granule prevents the entry of ribosomes, but the physical attributes of a condensate and its dynamic exchange with the surrounding cytoplasm seem to complicate such a static interpretation. Nevertheless, more contemporary microscopy techniques confirm that components of the ribosome, including 18S rRNA and ribosomal proteins RPL7a and RPL10, are excluded from the germ cell condensates [32,44,62]. The situation is more complicated for ribosome-associated factors that catalyze the mRNA recruitment and decoding, namely translation initiation factors. Many, including multiple paralogs of eIF4E, eIF4G, and eIF5B, are found to be concentrated at the periphery or in some cases sequestered within the germ granule [12,35,55,62,76,77,78,79,80]. Regardless of where in the transition from “inside” to “outside” of the condensate the first events of mRNA translation occur, it seems apparent from localization studies and translational control experiments that the condensates themselves serve a functional role in the post-transcriptional control programs on developmentally regulated mRNAs. However, several recent observations after genetic perturbations of granules have put back into question whether condensates per se enforce mRNA regulation (see below).

### 2.3. The Transition to Translational Activation

These LLPS condensates are dynamic structures that likely exchange both the protein and RNA content with the cytoplasm and adjacent granules but appear to exclude ribosomes [25,44,81,82]. Certain *C. elegans* eIF4E paralogs partition to granules, as well as to the soluble cytoplasm [37,55,83,84,85,86], similarly to observations in other species [12,54,62,87]. Translation factors eIF4E and eIF4G associate with the m7GpppN mRNA cap, along with the helicase eIF4A, to initiate their engagement with ribosomes (reviewed in [3,4,88,89,90,91]). Many studies demonstrate that eIF4 factors also play active roles, both positive and negative, in the mRNA translational control that is vital for germ cell differentiation in both plants and animals [1,2,3,5,69,88,92,93,94]. Their direct linkage to mRNA’s recruitment to ribosomes is critical because the synthesis of new proteins drives the differentiation of germ cells into gametes and subsequent embryos. As described above, the way mRNAs are handled during nuclear export sets the agenda for their translational control in the cytoplasm [24,55]. The implication of such constraints on mRNA life is that encounters during export are likely to involve eIF4 factors prior to their exposure to ribosomes. Studies in several animal germline and embryo settings have recently exposed that linkage both physically and functionally.

### 2.4. mRNP Remodeling During Stress and Aging: Altered Condensate Dynamics

Other LLPS condensates that carry mRNPs as cargo in specific cellular contexts, namely P bodies and stress granules (SGs), have also emerged as places of post-transcriptional regulation. Such structures feature prominently in the context of cellular stress and aging [95]. In somatic cells, the composition and assembly dynamics of P bodies and SGs in response to extrinsic stress such as heat shock, osmotic imbalance, and oxidative damage are well-characterized [96,97]. These conditions, as well as stress associated with natural aging, modify the biochemical and physical nature of P bodies and SGs in reproductive germ cells [82,98]. It is thought that P bodies serve as hubs of mRNA decay, while SGs serve as storage/sorting sites for translationally repressed messages [99]. In SGs, LLPS condensate formation is a consequence of the RNA-binding domains of resident RBPs interacting with recruited mRNAs [100]. However, there is some disagreement about the functional relevance of these macroscopic condensates [101]. It has been demonstrated that macroscopic P body formation per se is not required for mRNA decay, and some mRNAs that localize to P bodies in fact re-enter polysomes [102,103,104]. Broad evidence from yeast to mammals indicates that SG protein–protein interactions change slightly upon granule assembly [105,106]. In addition, certain yeast mutants defective in P body formation still assemble relevant cytoplasmic mRNPs, which are undetectable by light microscopy [107]. However, it remains likely that the condensation of these macroscopic structures enhances the plasticity and/or fidelity of post-transcriptional control programs. These may function globally or on certain transcripts under relevant contexts, such as in response to a particular stress or the coordination of developmental processes. Here too, the macroscopic condensates may enhance relevant functions because competing affinities of protein–protein interactions alter their recruitment of mRNAs [100].

Less is understood about how SGs and P bodies contribute to gene expression control programs in germ cells, gametes, and embryos. These cells already harbor specialized condensates/granules exemplified by *C. elegans* P granules, *Drosophila* polar granules, and mammalian chromatoid bodies [82,98,108]. Germline condensates are thought to safeguard germline RNAs, coordinate post-transcriptional programs, and preserve fertility during both extrinsic stressors, intrinsic challenges such as meiotic arrest, or even developmental timing cues [50,51]. For example, in *C. elegans* oocytes, heat shock or meiotic arrest cause the formation of large P-body-like condensates containing factors associated with polyadenylation, mRNA decay, and translational regulation (e.g., decapping enzyme DCP-2, CGH-1 helicase, poly(A) binding protein, and CAR-1 and TIAR-1 repressors). Such structures disassemble upon recovery from heat stress or the resumption of ovulation [109]. Yet their effects can be long lived, even cooperating over generations to suppress protein expression [50]. These assemblies are not physically homogeneous: PGL-1-associated condensates are highly dynamic and liquid-like, MEG-3 condensates are more gel-like and static, while MEX-3/CGH-1 condensates display intermediate properties [110]. Such phase heterogeneity may allow for the selective sequestration or release of certain mRNAs and RBPs in response to extrinsic or intrinsic cues, either globally or on specific mRNAs. Further complicating condensate heterogeneity is the observation that the ribosome engagement of an mRNA can effectively prevent recruitment to SGs, which may have a unique regulatory significance for those encoding upstream open reading frames (uORFs) [111]. Germline-specific condensates do interact (and perhaps subsume) P bodies or their components [50,51]. How they differ from their canonical counterparts in somatic cells, however, remains murky. Current models suggest that together germ granules, P bodies, and SGs help to “intelligently” distinguish transcripts to be translated, stored, or repressed during stress, while simultaneously maintaining germ-specific programs essential for fertilization and embryogenesis.

### 2.5. Proteins with Intrinsically Disordered Regions (IDRs) Promote Condensation

The physical interactions between mRNA and RBPs may approach critical concentrations inside cells, which leads to adsorption in a manner that is not truly solubilization [112,113]. These mixtures transition into LLPS condensates, sometimes referred to as membrane-less organelles [114]. Such condensates consist of sequence-specific RNA-binding proteins and IDR-containing sequence-non-specific RNA-binding proteins (Figure 1). In animal germ cells and embryos there is evidence that IDR proteins direct the assembly of RNA-rich germ granules, but it is unclear whether they recruit mRNAs directly or indirectly. For example, the *C. elegans* IDR proteins MEG-3/4 appear directly responsible for recruiting mRNAs to P granules via a kinetic trapping mechanism [33,112,115], while the *D. rerio* IDR-containing protein, Bucky ball (Buc), relies on the RBP Rbm24a to recruit mRNAs to germ granules [116,117]. Thus, there is a precedent for both direct and indirect mechanisms of mRNA recruitment to condensates, that the process is context- and species-dependent [101]. But the physical evidence for spontaneous RNA organization goes beyond germ granules [115,118,119]. In vitro experiments determined that concentrated RNA–IDR protein mixtures spontaneously induce phase separation, exhibiting different properties based on stoichiometry [113,120,121]. Arginine- and lysine-rich regions are common in general RBPs and have a propensity to be disordered, allowing them to readily phase separate [122]. The dynamics of such proteins when mixed with homopolymeric RNA [(e.g., poly(rC)] in vitro showed that they jointly form or partition into condensates [120]. LLPS occurs where RNA and proteins undergo adsorption and transition into condensates by the “wetting” of surface structures. Physical microscopy evidence in vitro as well as inside *C. elegans* embryos suggests that between two equally wettable phases, the RNA–protein complex localizes to a domain interface, similar to a Pickering emulsion [115]. In vivo it is clear that the self-organizing nature of the LLPS phenomenon is advantageous for germ cells and embryos to coordinate but spatially and temporally regulates their mRNA populations.

LLPS condensates are not unique to mRNA translation and stability. Large-scale condensates form on active chromatin DNA where a myriad of transcription factors are initiating a pioneer round transcription by RNA polymerase II [123]. Likewise, transcriptionally active nucleoli form regional condensates [118]. From the nucleolar example it was recognized that cooperation exists between DNA and sequence-specific proteins versus IDR proteins, driven by the miscibility of the nucleic acid and protein [118]. The study reports that the IDR of purified nucleolar protein can itself phase separate, whereas sequence-specific RNA-binding domains allow multiphase, layered separations [118,121]. The implication for cytoplasmic condensates is that repressive mRNPs include functional sequence-specific proteins and adhere to IDR-containing proteins to produce an organized suprastructure. It is important to note that condensates of the nucleus and cytoplasm are necessarily different both in their physical and biological nature. The former involves the nucleoplasm and chromatin enclosed within the nucleus. However, even bacterial transcription complexes not constrained by the nuclear envelope or chromatin nevertheless employ LLPS [119]. The physical nature of nucleic acid–protein condensates has been carefully discussed recently and will not be elaborated here [124]. An overall comparison to cytoplasmic mRNP-containing condensates may be helpful to understand biophysical forces at the heart of their formation and dynamics, with the caveat that their RNA–protein-only composition could have structural and biological relevance [101].

### 2.6. Stratified Arrangements of Protein and mRNA Within Condensates

An important physical property of LLPS condensates is their ability to create a “multilayered liquid” suprastructure for the positioning of their contents [121]. This organizing capacity may guide the dynamic movement of cargo mRNAs and proteins for non-random interactions relevant to functional gene expression. For example, *Drosophila* germ granule mRNAs were found to organize into homotypic clusters of single mRNA types (e.g., *Nanos*) visualized by single-molecule fluorescent in situ hybridization (smFISH) and super-resolution microscopy [125]. The authors postulated that by occupying specific positions as homotypic mRNAs within the granule, the non-distributed organization provides functional isolation, localization, and perhaps timing in their trafficking and eventual translation during embryogenesis. Strikingly, based on LLPS biophysics, the initial pre-wetting of the phase by the initially formed mRNP, in the presence of IDR proteins, leads to the additional recruitment of that mRNA sequence to the growing condensate. The effective crowding of *Nanos* and other regulated germline mRNAs by this self-sorting is thought to develop clusters that may regulate the timing from germ granules in a manner that is not strictly sequence-specific [125]. Such homotypic clusters can be induced in mammalian cells in cultures to the exclusion of non-targeted mRNAs using an engineered ArtiGranule scaffold IDR-RBP [126]. The designers observed mRNAs concentrating to a corona around the outer edge of the condensate formed and further proposed a causal relationship between the mRNA density and condensate size as well as layering. Perhaps most intriguing is that physiological conditions were able to prevent their formation or cause their dissolution. Although artificial, the in vivo observations strongly suggest native germline condensates as regulators of the mRNA spatial concentration and timed release.

The localization of germline- and embryonic-regulated mRNAs together in condensates/granules also positions them well for subsequent translation initiation. Careful imaging in Zebrafish embryos showed that the regulator RBP Deadend1 (DND1) actively moves *Nanos*-3 mRNA toward the germ granule periphery, where ribosomes accumulate, and away from the Vasa RNA helicase in the core [62]. Here *Nanos*-3 encounters translation initiation factors such as eIF4E and eIF4G, potentially setting up a directional protein synthesis mechanism that is engaged as soon as such mRNAs escape the condensate. Using single-molecule imaging in *Drosophila* embryos, *Nanos* mRNA was shown to remain in an mRNP structure with its 3’ UTR toward the “inside” as it emerges from the edge of the granule. At the cytoplasmic interface it is released by the translational repressor, Smaug, and engages eIF4G to begin recruitment to ribosomes [12]. We and others have shown that eIF4Es reside within the germ granules and take part in both the repressed mRNP and the eIF4-catalyzed engaged 48S initiation complex [37,53,55,78,80,127].

Two different eIF4E paralogs (IFE-1 and IFE-3) in *C. elegans* germ cells and embryos partition to germ granules as well as to soluble forms in the cytoplasm. The fraction retained in granules does so by binding their cognate eIF4E-interacting protein (4E-IP) and releases to cytoplasm when the 4EIP is no longer present [15]. IFE-1 binds specifically to the KH domain protein PGL-1 that first characterized worm “P granules” [37,84,85]. IFE-3 binds to the 4E transport protein IFET-1 (4E-T), which also serves as a translational repressor of mRNAs [35,55,78,128,129]. Their positioning within the condensate is also intriguing; IFE-1 and IFE-3 localize within the same granules but are separate and adjacent. High-resolution microscopy depicted IFE-3 at the lateral outer region, occasionally forming a halo around IFE-1 loci that are apical in the perinuclear granule [53]. Such substructures may indicate the type of homotypic clustering of eIF4Es and RBPs that is found for mRNAs in a condensate [125]. There also appears to be a hierarchy of stratification relative to other germ granule proteins. PGL-1 binds to both IFE-1 and GLH-1, which is the Vasa ortholog in *C. elegans*. PGL-1 has been reported to be more fluid in condensates compared to other granule residents [20,37,46,114]. The GLH-1 helicase appears as a raft-like structure toward the “inside” of perinuclear granules, resulting in a stratified substructure (Figure 1). *C. elegans* eIF4G (IFG-1) was found to be chiefly soluble in the germ cell cytoplasm, not concentrated at the granule periphery as in other species, but is still consistent with binding mRNAs shuttled by eIF4Es released from the granule [53]. The current model places repressed mRNA cap-binding mRNP entities that are facing outward and ready to engage the cap-dependent initiation mechanism [3,53,130,131,132,133]. Intriguingly, the *C. elegans* eIF4E paralogs demonstrate largely opposite translational activities on germ cell and embryonic mRNAs. IFE-1 actively enhances the translation of *pos-1*, *mex-1*, *mex-3*, *vab-1*, and other maternal mRNAs, while IFE-3 together with IFET-1 actively represses the translation of *pos-1*, *mex-3*, *fem-3*, and *fog-1*, as well as the *Nanos* mRNAs *nos-1* and *nos-2* [53,55,83,84]. Thus, while RBPs and helicases inhabit more centralized positions in the germ granule, the eIF4Es are localized to distinct positions in the corona, suggesting they may take part in the release of granule mRNPs to associate with eIF4G and be recruited to waiting ribosomes (see graphical abstract).

### 2.7. Condensates Collect Regulatory mRNPs but May Not Exert Regulation

What appears to be a directional sequence of binding events for mRNAs traversing perinuclear condensates made it easy to believe that the granule macrostructure was dictating mRNA regulation. In time IDR proteins that form structural networks to drive LLPS were found to be responsible for the biophysical attributes of *C. elegans* germ cell and embryo condensates [67]. In the most demonstrative case, intrinsically disordered MEG-3 protein was shown to recruit mRNAs to early embryonic granules in a sequence-non-specific fashion in P lineage blastomeres that represents the future germline [33]. In a series of elegant biochemical and in vivo studies, MEG-3 was shown to adsorb onto PGL-1 condensates reforming in the embryo posterior end [112,115]. To do so, MEG-3 forms a gel-like protective layer around a “liquid core” of PGL-1, thereby stabilizing P granules to trap cargo mRNAs by reducing the surface tension from the cytoplasm (Figure 1). Thus, the MEG-3 coating behaves like a Pickering agent for LLPS and recruits other components to help adhere P granules to the NPC. The genetic loss of MEG-3 caused a loss of granule association at NPCs as well as the visible disruption of the condensates in vivo [115]. Remarkably, however, the translational regulation and stability of MEG-3-associated mRNAs was unchanged by its loss [21]. These observations cast doubt on a causal link between mRNA repression and localization to the condensate. Instead, the new evidence points to RNA regulatory events (both mRNA and small RNAs) that are performed by the mRNP substructure, while spatial and organizational functions may be performed by the condensate suprastructure. Alternatively, it has been suggested based on biophysical behavior that condensates may form as a consequence of the regulation rather than as its cause [101].

What is now apparent is that the mRNPs that govern the translational regulation of these maternally/paternally stored mRNAs during germ cell differentiation and embryogenesis must be selective and dynamic. Our research discovered two *C. elegans* eIF4E paralogs, IFE-1 and IFE-3, that both localize within the condensates at all stages [55]. These eIF4Es are highly homologous, canonical mRNA cap-binding proteins that were thought to be redundant in the translation apparatus [37,86]. However, genetic analysis determined that IFE-1 promoted late spermatid maturation and oocyte ephrin signaling, whereas IFE-3 promoted the switch from sperm to oocyte differentiation, oocyte growth, and embryonic cleavage events [55]. Polysome analysis showed that each IFE regulates a different subset of mRNAs and does so by either the activation or repression of translation [55,64,84]. Within germ granules the eIF4Es are found in adjacent positions (described above), sequestered there by a cognate repressive 4EIP (IFE-1:PGL-1 and IFE-3:IFET-1). Immunoprecipitation and mass spectrometry (IP-MS) also showed that each paralogue binds to a distinct array of proteins, suggesting higher-order functional mRNPs [53]. Most extraordinary is that mRNAs found in IFE-3 mRNPs were found to be translationally recruited by IFE-1 to ribosomes [53,84]. We postulated that at some point IFE-3 shuttles these mRNAs to the IFE-1 mRNP, either within the condensate or upon remodeling of the IFE-1 mRNP when it engages IFG-1 (eIF4G) upon entering the cytoplasm to engage with ribosomes [15,53]. The proposed cap-bound “hand-off” would require opposing yet cooperative interactions between two eIF4E isoform mRNPs that are also supported by the IP-MS data. Together with translational control data, these findings imply dynamic eIF4E mRNP subtype remodeling during development. Multiple eIF4E paralogs exist in all eukaryotes [94,134] and are likely to have distinct roles in germline and embryo translational activation. However, their unique functions and juxtaposition in germ granules are less well described.

Despite the functional evidence for singular active and repressed mRNP complexes for the eIF4E paralogs, there is compelling structural evidence that each one forms alternative repressed and active protein–protein contacts. Such alternative binding events create positive and negative translation complexes that bind mRNA caps in both cases [15]. Co-crystal structures of human and *Drosophila* eIF4E complexed with eIF4G- (positive) and 4E-IP- (negative) binding peptides show that mutually exclusive interactions form in translational repression versus activation [135,136]. Directly orthologous structures are predicted using AlphaFold3 for the *C. elegans* canonical eIF4E (IFE-3) with eIF4G (IFG-1) and the 4E-T ortholog (IFET-1; Figure 2). Interestingly, suppressed 4E-T expression gives rise to phenotypes ranging from the full loss of oogenesis to ovarian insufficiency in worms, frogs, and mice [35,79,137,138]. Even modest mutations in the human gene encoding 4E-T (EIF4ENIF1) result in premature ovarian insufficiency in patients [139]. In all cases, the function of the 4E-T mRNP complexes is to repress the translation of mRNAs that promote meiotic progression specifically for oogenesis [35,55,140]. Therefore, a simple switch from eIF4E interactions in the repressed 4E-T complex to those of the eIF4E-eIF4G translation initiation complex may be the basis of mRNP remodeling to bring about a change in translational control (Figure 2). The condensate interface may provide an environment conducive to the protein synthesis switches that set undetermined germ cell progenitors on the course to oocyte and embryo development.

## 3. Conclusions

### 3.1. A New Way to Envision mRNA–Protein Dynamics in the Cytoplasm

In all that has been learned about mRNAs and how they assemble into mRNPs, there is something radically different in how they engage LLPS condensates. mRNPs use them to potentially remodel during the mRNA lifetimes and follow a designated path from the nucleus to ribosome. In that dynamic experience they experience repression, oligomerization, protein partner swapping, release into the cytoplasm, and the engagement of either the ribosome or degradative P bodies. The result is the spatial and temporal control of their expression that has been a hallmark of oogenesis, spermatogenesis, and early embryogenesis. The spatial–temporal arrangement and storage in germ granules may compensate for a lack of transcriptional regulation found in somatic cell types. Instead, mRNAs segregated and localized on the corona of physically malleable condensates with translation factors prepositioned and ribosomes just outside the boundaries provide the conditions that inform the overall mechanism and specificity of steps in mRNA regulation implied by the members.

There are three anecdotal facts that are consistent with a transition from negative to positive mRNA translational control that occurs on the periphery of germ granules: (1) Active cap-binding translation factors eIF4E (and eIF4G in some species) are concentrated in the external layers of the condensate [12,53,62]. Some of the factors are cell-specific paralogs that are known to have special translational roles for the regulated mRNA subpopulation. (2) Integral ribosome components do not appear to permeate the condensate body but do concentrate around the periphery [44,62]. (3) Physical measurements suggest that mRNA is less structured near the condensate periphery than internally, suggesting it may be unwound, perhaps by eIF4A or Vasa helicase, in preparation for active translation [44,53,126,140,142].

### 3.2. Emerging Technologies That May Address Dynamics and Causality

At present knowing whether translational repression-to-activation switches occur within, outside, or at the periphery of germ granule condensates remains a major challenge. Emerging single-molecule methodologies are proving helpful to establish how such transitions relate to subgranule organization and the individual mRNP architecture. Improved approaches such as MS2/MS2 coat protein labeling of transcripts [143,144] combined with nascent translation reporters like SunTag/MoonTag [145] enable the visualization of individual mRNAs and their spatiotemporal translation in real time. Indeed, these combined techniques detected an orientation of *Nanos* mRNA in granules [12]. The imaging suggested that initiation events occur on *Nanos* mRNA while the 5′ cap is leaving the condensate and the 3′ poly(A) tail is still buried inside. Yet with each advance comes further uncertainty. High-resolution imaging suggests that the precise boundary between the “inside” and “outside” of the condensate may be difficult to define in the context of productive translation initiation. Future studies are likely to couple such imaging techniques with precise genetic perturbations, such as CRISPR-mediated site-directed mutagenesis of genes encoding condensate scaffolding proteins or the disruption of specific transcript sequence motifs. In combination with biochemical studies, such tools could provide key insights into how subgranule structures relate to dynamic transitions between translational repression and the activation of germ cell mRNPs.

## Figures and Tables

**Figure 1 biology-14-01430-f001:**
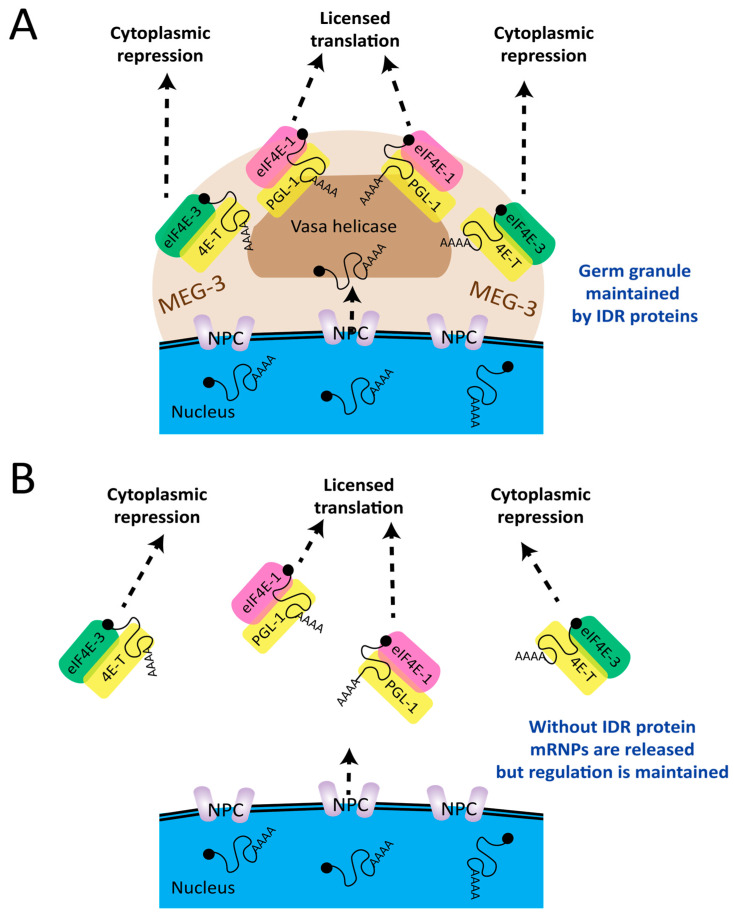
*Intrinsically disordered regions (IDRs) provide cohesion to condensates.* In this example, germ cell and embryo condensates are macrostructures formed by physical interaction of IDR-containing proteins, such as MEG-3, with RNAs. The IDR proteins have no known sequence specificity but rather associate biophysically to promoting liquid–liquid phase separation (LLPS). (**A**) mRNAs emerge through the nuclear pore complex (NPC) to be concentrated with IDR proteins after unwinding by helicases such as Vasa/GLH-1. They assemble mRNP substructures as they bind RNA-binding proteins (RBPs, e.g., 4E-T and PGL-1, yellow). Some RBPs are also eIF4E-interacting proteins (4E-IPs), which recruit cap-binding proteins eIF4E-1 and eIF4E-3. Since cap-binding proteins show no sequence specificity, the mRNP identity is set by the RBPs on each mRNA. It is the nature of each mRNP that dictates their fate when they encounter cytoplasm (dotted arrows). (**B**) Genetic depletion of the IDR protein MEG-3 dissolves the condensate macrostructure but leaves the mRNP substructures intact. Both translational control and mRNA stability of the cargo mRNAs was observed to be maintained despite the loss of germ granules (see Scholl et al. 2024) [21].

**Figure 2 biology-14-01430-f002:**
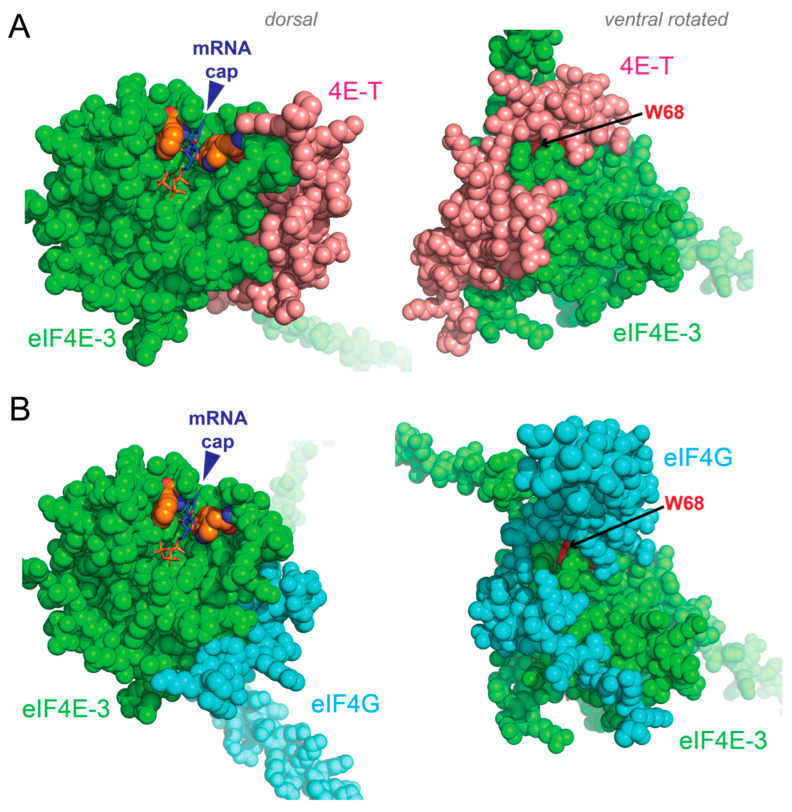
*Models indicate that activating and repressing proteins bind eIF4Es similarly, but not identically, on the dorsal face.* Alternative structures of canonical *C. elegans* eIF4E-3 bound to peptides from its two partners eIF4E-T (4E-T) and eIF4G. (**A**) The cap-binding pocket (blue arrowhead) and access to the mRNA 5’ end are unobstructed by the protein-binding partners. The m7GTP cap (stick structure) is trapped by pi orbital stacking from two tryptophan residues (orange) and electrostatic interactions on the pocket floor with the triphosphate linkage. There is interaction with only the initial two nucleotides (not displayed) of the mRNA. (**B**) Both 4E-T (magenta) and eIF4G (light blue) interact with the dorsal face of eIF4Es based on co-crystal structures of human and *Drosophila* duplexes [135]. A single tryptophan (buried; equivalent to W68 in eIF4E-3) is indispensable for interaction with both partners. However, the surface structures indicate that eIF4E interacts differently with eIF4G or 4E-T. Translational recruitment may involve remodeling of the eIF4E mRNPs by swapping those interactions. Structural interactions were predicted by AlphaFold3 [141] and represented using PyMol 3.1.5.1.

## Data Availability

No new data were created or analyzed in this study. Data sharing is not applicable.

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
