# Peer review of "Remodeling of Germ Cell mRNPs for Translational Control"

_biology, 2025, doi:10.3390/biology14101430_

Round 1

Reviewer 1 Report

Comments and Suggestions for Authors

Keiper and Huggins provide a timely and informative overview of translational control of mRNAs within germ granules. The review succinctly traces the history of germ granules, emphasizes that translation tends to occur at condensate borders—implicating subgranular mRNA dynamics in regulatory outcomes—and compares germ granules with stress granules and P bodies. The discussion of RNA–IDR protein interactions in condensate formation and the roles of eIF4E paralogs in controlling translation of condensed mRNAs is valuable. Overall, the manuscript is well written and will be useful to researchers studying mRNA regulation in phase-separated compartments. Several revisions would strengthen clarity, cohesion, and balance.

Major Comments

  1. Section 2.5 currently reads as relatively independent from 2.4 and 2.6. Consider integrating the discussion of IDR protein–driven condensation into Section 2.1, where you introduce why condensates form. This will improve conceptual continuity from formation mechanisms to functional outcomes.
  2. In the graphical abstract, explicitly label the blue ellipse as the nucleus.

Reconsider the orientation of mRNAs across figures to align with lines 299–301, where you note that in Drosophila the nanos mRNA 3′UTR faces the “inside.” Ensure this convention is used consistently.

Provide concise, informative titles in the legends of Figures 1 and 2.

  1. The review would benefit from a forward-looking perspective. Please highlight emerging tools to interrogate mRNA–protein dynamics in germ cells, such as: targeted protein degradation (e.g., AID, dTAG), optogenetic/chemogenetic control of condensates, live mRNA imaging (MS2/MCP, PP7/PCP, SunTag/MoonTag, CRISPR-based RNA imaging). Briefly outline how these approaches can resolve subgranular dynamics and causality.
  2. In Section 2.5 (lines 235–236), you state that “In animal germ cells there is evidence that the IDR proteins themselves recruit mRNAs to allow assembly of RNA-rich germ granules.” This appears context- and species-dependent. For example, in fish, Buc (an IDR protein) forms condensates, but recruitment of mRNAs is mediated by a Buc-binding RBP; there is also evidence that RNA may not drive condensation but instead acts as a client molecule. Please temper this statement, present both models, and reconcile the discrepancy with appropriate citations.

Minor Comments

  1. Language and readability

Several sentences are difficult to follow; please revise for concision and clarity at lines 19–21, 314–316, 335–339, and 354–356.

  1. Figure attributions

In the legend of Figure 2, properly cite AlphaFold3 (include the correct reference and version information).

  1. Spelling, nomenclature, and formatting

Line 41: “like” can be deleted.

Page 4, first sentence of the second paragraph: “translation” should be “translated”.

Gene/mRNA nomenclature: many mRNA names should be lowercased and italicized on pages 4, 7, and 8, following field conventions.

Line 153: add the missing parenthesis.

Line 294: “RPB” should be “RBP.”

Author Response

There were many encouraging and supporting comments by the reviewers, in addition to the critiques. We would like to thank all 3 reviewers and let them know that their enthusiasm and positive approach are heartening.

Comments and Suggestions for Authors

Comment 1: Several revisions would strengthen clarity, cohesion, and balance.
Response 1: We thank the reviewer for the alternate perspectives in suggested revisions. She/he correctly suggests these revisions provide more balance.

Major Comments
Comment 2: Section 2.5 currently reads as relatively independent from 2.4 and 2.6. Consider integrating the discussion of IDR protein–driven condensation into Section 2.1, where you introduce why condensates form. This will improve conceptual continuity from formation mechanisms to functional outcomes.
Response 2: We also recognize the connection made by the reviewer and agree it would be feasible to integrate 2.5 into 2.1. However, it was felt that the section 2.1 is already somewhat cumbersome because it attempts to establish the entire flow of mRNA experience. The character of IDR protein-driven condensation seemed to significant enough to merit a separate section. We preferred not to disrupt the flow of germline mRNA experience to first encounter ribosomes (2.2), then transition to translational activation (2.3), then changes with stress (2.4) before diving deep into the physical attributes of IDR proteins (2.5). While it could likely be restructured, we will maintain our progression

Comment 3: In the graphical abstract, explicitly label the blue ellipse as the nucleus.
Response 3: We relabeled the blue ellipse as nucleus.

Comment 4: Reconsider the orientation of mRNAs across figures to align with lines 299–301, where you note that in Drosophila the nanos mRNA 3′UTR faces the “inside.” Ensure this convention is used consistently.
Response 4: We truly appreciate the insightful observation by the reviewer about our figures to better relate them to the text. We have oriented the mRNAs “inward” in both the Graphical Abstract and in Fig. 1. We also reversed the red “repression” lines to signify inhibition by the mRNPs, not the ribosomes.

Comment 5: Provide concise, informative titles in the legends of Figures 1 and 2.
Response 5: Concise title in italics were added to all figures (pp. 2, 4, 10).

Comment 6: The review would benefit from a forward-looking perspective. Please highlight emerging tools to interrogate mRNA–protein dynamics in germ cells….Briefly outline how these approaches can resolve subgranular dynamics and causality.
Response 6: At the reviewer’s request, a new subsection (3.2 Emerging technologies that may address dynamics and causality) was added to the end of Conclusions specifically dedicated to emerging tools and where they will enhance the field.

Comment 7: In Section 2.5 (lines 235–236), you state that “In animal germ cells there is evidence that the IDR proteins themselves recruit mRNAs to allow assembly of RNA-rich germ granules.” This appears context- and species-dependent…..Please temper this statement, present both models, and reconcile the discrepancy with appropriate citations.
Response 7: The zebrafish examples of actions by buckeyballs and Rbm24a provide a very useful contrast to that of C elegans MEG-3/4. Both are now described and the caveat of context and species added (p. 7, Section 2.5). We also tempered the wording in the Abstract (line 31) from “…assumptions…were misguided…” to “assumptions….may have been misguided….”

Minor Comments
    Language and readability
Comment 8: Several sentences are difficult to follow; please revise for concision and clarity at lines 19–21, 314–316, 335–339, and 354–356.
Response 8: For clarity, run-on sentences have been broken and simpler wording used in each of these places.

Comment 9: Figure attributions. In the legend of Figure 2, properly cite AlphaFold3 (include the correct reference and version information).
Response 9: The reference (Abramson, 2024) for AlphaFold3 is now given.

    Spelling, nomenclature, and formatting
Comments 10-14: Line 41: “like” can be deleted.
Page 4, first sentence of the second paragraph: “translation” should be “translated”.
Gene/mRNA nomenclature: many mRNA names should be lowercased and italicized on pages 4, 7, and 8, following field conventions.
Line 153: add the missing parenthesis.
Line 294: “RPB” should be “RBP.”
Response 10-14: Typographical and mRNA/protein name errors were corrected to reflect literature conventions/rules (https://www.ncbi.nlm.nih.gov/genbank/internatprot_nomenguide/).

Reviewer 2 Report

Comments and Suggestions for Authors

This article provides a deep insight of recent findings on mRNP remodeling and germ granule biology. It covers very well the existing literature in depth, with critical reasoning and analysis suited and establishing connections between findings.

In several instances (e.g., lines 334–356), the authors conclude that condensates might form as a consequence rather than regulatory by itself. While intriguing, this statement could overreach the presented evidences. I suggest,  it might be more appropriate to frame such bold statements as a sort of 'emerging hypothesis'  in the field and warrants further investigation.

The Figure 1 is highly cluttered, could be improved by giving a clear idea by including active vs repressive modules for simplicity.

A schematic of future directions and questions in this field would be nice addition in the conclusion section.

Should include a keywords/definition schema/module for terms commonly used like “granules,” “condensates,” “macrostructures,” for better readability. 

Comments on the Quality of English Language

There are several awkward phrasing, and incoherent sentences throughout the article. The manuscript should be professionally proofread to meet the standards of scientific writing.

For example- "This article will focus on the role of mRNPs in germ cells and embryos; the work on neuronal mRNP translational control has been reviewed elsewhere [9,22]. Similarly, there are numerous reviews that give more comprehensive literature history of germ cell and embryo RBPs, mRNA repression, small RNA regulation (miRNAs, endo-siRNAs, etc.), cytoplasmic poly(A) elongation, and mRNA turnover that is not possible to fully address here (reviewed in [5,14,15,17,23-29]. "

"Those proteins nucleate a “micro-scale” RNA-protein complex, defined for the sake of this review as mRNPs."

Author Response

There were many encouraging and supporting comments by the reviewers, in addition to the critiques. We would like to thank all 3 reviewers and let them know that their enthusiasm and positive approach are heartening.

Comments and Suggestions for Authors
Comment 1: In several instances (e.g., lines 334–356), the authors conclude that condensates might form as a consequence rather than regulatory by itself. While intriguing, this statement could overreach the presented evidences. I suggest,  it might be more appropriate to frame such bold statements as a sort of 'emerging hypothesis' in the field and warrants further investigation.
Response 1: The reviewer is correct that the “consequence” interpretation is still hypothetical. For that reason the subtitle 2.7 is carefully worded to say, “…may not exert regulation”. At the end of this section we revisit the known observations, and propose the regulatory/organizational functions of condensates as a primary view and the “consequence” model as an alternative. Wording at the beginning and end of section 2.7 have been changed to make the alternatives clearer.

Comment 2: The Figure 1 is highly cluttered, could be improved by giving a clear idea by including active vs repressive modules for simplicity.
Response 2: The images in Figure 1 were simplified and excessive wording removed for clarity. Modules were corrected to orient mRNAs toward the condensate core as described in the text.

Comment 3: A schematic of future directions and questions in this field would be nice addition in the conclusion section.
Response 3: To address future directions, a new subsection (3.2 Emerging technologies that may address dynamics and causality) was added to the end of Conclusions.

Comment 4: Should include a keywords/definition schema/module for terms commonly used like “granules,” “condensates,” “macrostructures,” for better readability. 
Response 4: The terms listed here have now all been included in the Keywords section. While the journal does not provide a format for definitions and limits figures, we sought to explain these terms within the text.

Comments on the Quality of English Language
Comment 5: There are several awkward phrasing, and incoherent sentences throughout the article. The manuscript should be professionally proofread to meet the standards of scientific writing.
For example- "This article will focus on the role of mRNPs …..address here (reviewed in [5,14,15,17,23-29]. "
"Those proteins nucleate a “micro-scale” RNA-protein complex, defined for the sake of this review as mRNPs."
Response 5: The reviewer rightly points out awkward wording and excessively complicated text. For clarity in the sections quoted above, we have deleted unnecessary statements, broken up run-on sentences and provided simpler wording used in each of these places.

Reviewer 3 Report

Comments and Suggestions for Authors

Presented review covers very important topic of translational regulation by mRNP condensates. It is really interesting and well written. May be helpful for many scientists, both - involved in condensates study, as well as ones who want to expand their knowledge even if not study condensates themselves.

I have only few minor comments:

  1. Check the names of all proteins and genes/RNAs and follow general guidelines (e.g. human - uppercase, mammals first letter upper, then lowercase etc.., genes and mRNA - italic...)
  2. In line 56-57 you mentioned differentiation "without substantial transcriptional activation", is substantial transcriptional repression observed?
  3. Statement in line 141-145 "Furthermore, the physical characteristics of cyto-
    plasm are also non-homogeneous, especially in early embryos that must establish polarity for future developmental axes. The landscape outside of the granules therefore further influences mRNA translational control spatially in various blastomeres for cell-specific translation of maternal mRNAs" is a little bit complicated and unclear for me, can you clarify heterogenity of cytoplasmic landscape outside of the granules in various blastomeres.

Author Response

There were many encouraging and supporting comments by the reviewers, in addition to the critiques. We would like to thank all 3 reviewers and let them know that their enthusiasm and positive approach are heartening.

Comments and Suggestions for Authors
Comment 1: I have only few minor comments:
Check the names of all proteins and genes/RNAs and follow general guidelines (e.g. human - uppercase, mammals first letter upper, then lowercase etc.., genes and mRNA - italic...)
Response 1: There were errors and typos in our mRNA/protein names. We looked up the species nomenclature rules (https://www.ncbi.nlm.nih.gov/genbank/internatprot_nomenguide/) and corrected them to reflect literature conventions.

Comment 2: In line 56-57 you mentioned differentiation "without substantial transcriptional activation", is substantial transcriptional repression observed?
Response 2: The very old papers cited here (Kimelman, etal. as well as Newport and Kirschner) from Xenopus MBT embryos showed rather conclusively that transcriptional suppression beginning in late meiosis is maintained for numerous cell cycles following fertilization, although the duration is variable among species. We felt it would be distracting to say more.

Comment 3: Statement in line 141-145 "Furthermore, the physical characteristics of cytoplasm are also non-homogeneous, especially in early embryos… is a little bit complicated and unclear for me, can you clarify heterogenity of cytoplasmic landscape outside of the granules in various blastomeres.
Response 3: Admittedly, the language in this section became a bit flowery and vague. Here too, breaking up sentences and using more concrete terms and examples have hopefully clarified the meaning.